# BRAIDing receptors for cell-specific targeting

**Hui Chen[†], Sung-Jin Lee[†], Ryan Li, Asmiti Sura, Nicholas Suen, Archana Dilip, Yan Pomogov, Meghah Vuppalapaty, Timothy T Suen, Chenggang Lu, Yorick Post, Yang Li***

Surrozen Inc, South San Francisco, United States

**Abstract** Systemic toxicity is a major challenge in the development of therapeutics. Consequently, cell-type-specific targeting is needed to improve on-target efficacy while reducing off-target toxicity. Here, we describe a cell-targeting system we have termed BRAID (BRidged Activation by Intra/intermolecular Division) whereby an active molecule is divided into two inactive or less active parts that are subsequently brought together via a so-called 'bridging receptor' on the target cell. This concept was validated using the WNT/β-catenin signaling system, demonstrating that a multivalent WNT agonist molecule divided into two inactive components assembled from different epitopes via the hepatocyte receptor βKlotho induces signaling specifically on hepatocytes. These data provide proof of concept for this cell-specific targeting strategy, and in principle, this may also allow activation of multiple signaling pathways where desirable. This approach has broad application potential for other receptor systems.

**\*For correspondence:**
yang@surrozen.com

[†]These authors contributed equally to this work

## eLife assessment

This **important** study presents a new way to selectively activate a cell signaling pathway in a specific cell type by designer ligands that link signaling co-receptors to a marker specific to the target cells. **Convincing** experimental results demonstrate that the agonist molecules activate Wnt signaling in target cells expressing the marker as intended. More broadly, this concept could be used to induce Wnt signaling or another pathway initiated by co-receptor association in a cell type-specific manner. In vitro results in this study could be further strengthened by assessing the biological consequences of Wnt activation in target cells.

## Introduction

Receptor-mediated signaling is fundamental to cell–cell communication, regulation of cellular growth/differentiation, maintenance of tissue function/homeostasis, and regulation of injury repair. When a cell-specific response is necessary, nature achieves that selectivity through various mechanisms such as cell-specific receptors, ligand gradient, short-range ligands, and direct cell–cell contacts. However, most ligand receptor systems have pleiotropic effects due to the broad expressions of receptors on multiple cell types. While this may be important when a coordinated response in multiple cell types or tissues is needed, achieving cell-specific signaling to avoid systemic toxicity or off-target effects remains a major challenge in both research and therapeutic development. In addition, it is often difficult to identify differences between normal and diseased cells for a particular signaling pathway of interest, making it challenging, if not impossible, to modulate that signaling pathway selectively in either diseased or normal cells. Therefore, methods for effective cell targeting are needed for both research and therapeutic development.

Compared with cell-targeted antagonists, cell-targeted agonists are more complex to design. In addition to cell targeting, agonist molecules are also subject to stringent requirements in terms of affinity, epitope, and geometry (*Dickopf et al., 2020*). Previous efforts to engineer cell-selective growth factors and cytokines have employed various approaches, such as the 'chimeric activators' concept (*Cironi et al., 2008*), where a cell-targeting arm ('targeting element') is attached to either the wild-type ligand or a mutated ligand with reduced affinity toward the signaling receptor ('activity element'). Such an approach has been applied to several signaling pathways, such as those based on erythropoietin (*Taylor et al., 2010*; *Burrill et al., 2016*), interferons (*Cironi et al., 2008*; *Garcin et al., 2014*), and interleukin-2 (*Ghasemi et al., 2016*; *Lazear et al., 2017*). While some selectivity has been achieved by attaching a targeting arm to a wild-type ligand, up to 1000-fold selectivity has been achieved using a mutated ligand (*Garcin et al., 2014*).

The 'chimeric activators' approach is based on the cooperativity concept, where mutations that weaken the affinity of a natural ligand to its receptor are selected as the 'activity element.' Due to this weakened affinity, the mutant 'activity element' alone displays a significantly right-shifted dose–response curve (i.e., lower potency) in an activity assay, for both target and non-target cells that express the signaling receptors. When the 'targeting element' is tethered to the mutant 'activity element,' the 'targeting element' helps increase the local concentration of the mutant 'activity element' on the desired target cell surface. This combination drives engagement of the signaling receptor and subsequent intracellular signaling activation, left-shifting the dose–response curve (i.e., higher potency) on the target cell and creating a separation between target vs. non-target cells. While conceptually elegant, identifying the appropriate mutations that achieve the precise reduction of affinity to be rescued by the 'targeting element' is not trivial. In some cases, while the potency could be rescued, the maximal signaling strength remained compromised, resulting in a partial agonist. In addition, the selectivity is not exquisite, with most literature examples reporting a modest tenfold separation between target vs. non-target cells. Additional strategies that show promising results have also been reported; for example, inactive pro-drugs that require tumor-associated proteases for activation (*Puskas et al., 2011*; *Skrombolas et al., 2019*); complementation of two inactive components via targeted assembly in trans (*Banaszek et al., 2019*) and passive enrichment on the target cell leading to colocalization of monomeric subunits (*Mock et al., 2020*) or inactive 'split' components (*Quijano-Rubio et al., 2023*).

Here, we present a novel concept to achieve cell targeting for ligand/receptor systems that involves multicomponent receptor complexes, where a productive signaling competent receptor complex is directly assembled through a 'bridging element.' We have tested this concept using the WNT/β-catenin signaling pathway as a model system.

The WNT pathway is highly conserved across species and crucial for embryonic development as well as adult tissue homeostasis and regeneration (*Nusse and Clevers, 2017*). WNT-induced signaling through β-catenin stabilization has been widely studied and is achieved by the ligand binding to frizzled (FZD) and the low-density lipoprotein receptor-related protein (LRP) families of receptors. There are 19 mammalian WNTs, 10 FZDs (FZD$_{1-10}$), and 2 LRPs (LRP5 and LRP6). WNTs are highly hydrophobic due to lipidation, which is required for them to function, and they are promiscuous, capable of binding and activating multiple FZD and LRP pairs (*Janda et al., 2012*; *Kadowaki et al., 1996*; *Dijksterhuis et al., 2015*). Elucidating the functions of individual FZDs in tissues has been hampered by difficulties in producing the ligands and their lack of receptor and tissue selectivity. Recent breakthroughs in the development of WNT-mimetic molecules have largely resolved the production and receptor-specificity challenges (*Janda et al., 2017*; *Chen et al., 2020*; *Tao et al., 2019*; *Miao et al., 2020*). Although tissue selectivity has been partly achieved by tissue injury (damaged tissues seem more sensitive to WNTs; *Xie et al., 2022*), the ability to target WNTs to specific cells and tissues would be a significant technical and therapeutic advancement.

The WNT mimetics reported so far are all bispecific and can simultaneously bind to FZDs and LRPs, and their optimal stoichiometry (at least for the antibody-based molecules) are tetravalent bispecific (2:2 format), requiring two FZD binders and two LRP binders in the same molecule to achieve efficient signaling (*Tao et al., 2019*; *Chen et al., 2020*). We have taken advantage of the fact that two FZD binders alone and two LRP binders alone do not signal. Cell specificity may be achieved by attaching a 'targeting element' (capable of binding to another cell surface receptor, called a bridging receptor here) to these two inactive molecules. Signaling-competent receptor complexes consisting of two

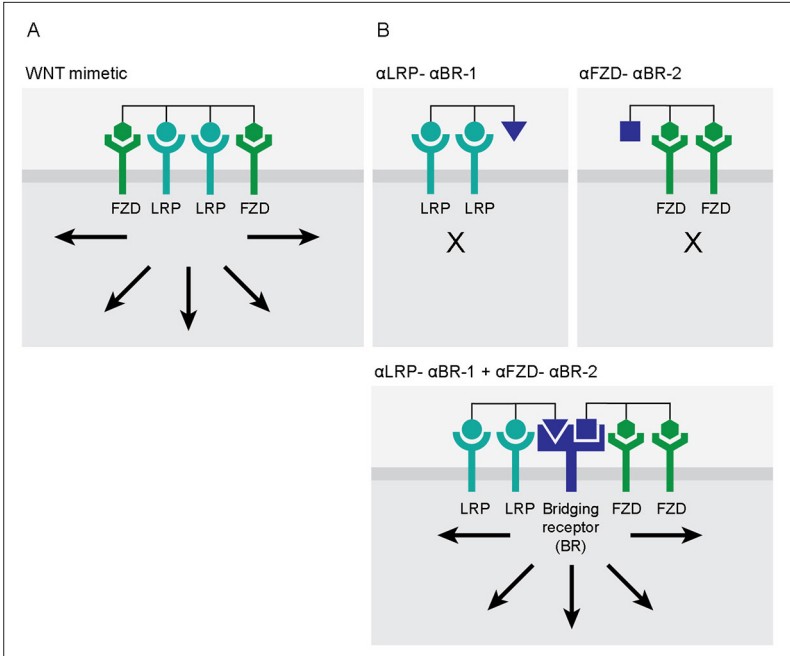

**Figure 1.** Activation of a signaling pathway by split molecules through binding to a bridging receptor on the target cell surface. (**A**) Efficient WNT/β-catenin signaling requires two frizzled (FZD)-binding domains and two lipoprotein receptor-related protein (LRP)-binding domains in one WNT mimetic molecule. (**B**) WNT mimetic molecule is split into one molecule having two FZD-binding arms (2:0) and a second molecule having two LRP-binding arms (0:2) and then each tethered to a different 'bridging element' binding to a different epitope on the same bridging receptor (named αLRP-αBR-1 and αFZD-αBR-2) (top two panels). On target cells where the bridging receptor is expressed, FZD and LRP are then assembled by αLRP-αBR-1 and αFZD-αBR-2 via the bridging receptor, recreating the signaling-competent receptor complexes (bottom panel). Arrows indicate activation of WNT/β-catenin signaling.

FZDs and two LRPs could then be assembled via the bridging receptor on the target cell surface. This approach reduces or eliminates the need to mutate and reduce the affinity of the 'active elements' toward the signaling receptor, and creates a highly cell-specific activation of the signaling pathway as the individual components are inactive. A detailed proof-of-concept study based on the WNT/β-catenin signaling pathway is presented herein. We have termed the use of cell-targeted WNT mimetics SWIFT (**S**plitting of **W**NT to **I**nduce **F**unctional **T**argeting).

## Results
### Conceptual design of cell-targeted activators via a bridging receptor
*Figure 1A* shows one optimized design for a WNT mimetic that is a tetravalent bispecific antibody-based molecule. It is important to note that efficient WNT/β-catenin signaling requires two FZD-binding domains and two LRP-binding domains in one molecule (*Chen et al., 2020*). *Figure 1B* shows a cell-targeting approach using the bridging receptor concept, that is, the SWIFT approach. The first step of this concept is to split the active molecule into inactive components. In the case of the tetravalent bispecific WNT mimetic, the active molecule is split into one molecule having two FZD-binding arms and a second molecule having two LRP-binding arms. These two inactive components are then each tethered to a different 'bridging element' binding to a different epitope on the same bridging receptor (the tethered molecules named αLRP-αBR-1 and αFZD-αBR-2 in *Figure 1B*, top two panels). αLRP-αBR-1 and αFZD-αBR-2 are each inactive on non-targeting cells where the bridging receptor is not expressed (*Figure 1B*, top panels). On targeting cells where the bridging receptor is expressed, FZD and LRP are then assembled or brought into proximity with one another by αLRP-αBR-1 and αFZD-αBR-2 via the bridging receptor, recreating the signaling-competent receptor complex (*Figure 1B*, bottom panel).

## Proof of concept with the targeted WNT mimetic molecules

To test the concept shown in *Figure 1B*, we selected the βKlotho and endocrine fibroblast growth factor 21 (FGF21) ligand system as the bridging receptor system. FGF21 is an endocrine hormone produced by the liver that regulates metabolic homeostasis (*BonDurant and Potthoff, 2018*). FGF21 signals through FGFR1c, FGFR2c, and FGFR3c in the presence of co-receptor βKlotho (*Zhang and Li, 2015*). The binding of FGF21 to the receptor complex is primarily driven by its affinity toward βKlotho via its C-terminal domain (*Lee et al., 2018*; *Shi et al., 2018*), while its N-terminal domain is important for FGFR interaction and signaling (*Yie et al., 2012*; *Micanovic et al., 2009*). βKlotho-binding antibodies that could induce βKlotho/FGFR signaling have also been identified, and one particular agonistic βKlotho antibody binds to a different epitope on βKlotho from FGF21 and does not compete with FGF21 binding (*Min et al., 2018*). Therefore, the following bridging receptor (βKlotho)-binding elements were selected to test the cell-targeting SWIFT concept:

- FGF21FL (full-length FGF21) that can bind to βKlotho and induces FGFR signaling.
- FGF21ΔC (FGF21 without the C-terminal βKlotho interaction domain) that does not bind βKlotho nor is capable of inducing FGFR signaling.
- FGF21ΔN (FGF21 without the N-terminal FGFR interaction domain) that binds βKlotho but does not signal.
- FGF21ΔNΔC that cannot bind βKlotho and cannot signal.
- 39F7 IgG that binds βKlotho, to a different epitope from FGF21, and can induce FGFR signaling.

The FZD- and LRP-binding domains selected were F (binds $FZD_{1,2,5,7,8}$) and L (binds LRP6), previously termed F1 and L2, respectively (*Chen et al., 2020*). Graphic representations of the binders and their various combinations are shown in *Figure 2A–C*.

To test the concept in *Figure 1B*, the FZD binder (F) was combined with two versions of the bridging receptor (βKlotho) binder, F-FGF21FL and F-FGF21ΔN (*Figure 2B*), and the LRP binder (L) was combined with the other bridging receptor (βKlotho) binder 39F7 as L-39F7 (*Figure 2B*).

We first confirmed the binding of the various fusion molecules shown in *Figure 2B and C* to their respective target proteins. All F- and L-containing molecules bind to either $FZD_7$ or LRP6E3E4 fragments as expected (*Figure 2D–J*). FGF21FL, FGF21ΔN, and 39F7 bind to βKlotho, while FGF21 without its C-terminal domain (FGF21ΔC and FGF21ΔNΔC) does not bind to βKlotho (*Figure 2D–J*). Therefore, the molecule formats of the various fusion proteins have no significant impact on the individual element's binding to their target receptors.

Next, we performed Octet-binding assays to assess whether F-FGF21 or L-39F7 allow simultaneous bindings to their target receptors. Sequential additions of the various F-FGF21 fusion molecules to the sensor surface, followed by $FZD_7$ CRD and then βKlotho show that a stepwise increase in binding signal is observed with $FZD_7$ CRD, and that F-FGF21FL and F-FGF21ΔN show additional binding to βKlotho but not F-FGF21ΔC nor F-FGF21ΔNΔC (*Figure 2K*). This suggests that F-FGF21FL or F-FGF21ΔN are capable of simultaneous binding to both $FZD_7$ and βKlotho receptors.

Sequential additions of the various L, 39F7, and αGFP (a negative control antibody) combinations to the sensor surface followed by LRP6E3E4 and then βKlotho revealed a stepwise increase in binding signal with L-39F7 but not the other negative control molecules, αGFP-39F7 and L-αGFP (*Figure 2L*). This suggests that L-39F7 can simultaneously bind to both LRP6 and βKlotho receptors.

The ability of this set of molecules to activate WNT/β-catenin signaling was assessed in WNT-responsive Huh7 and HEK293 Super TOP-FLASH (STF) reporter cells (*Zhang et al., 2020*). As shown in *Figure 3B*, the combination of F-FGF21FL or F-FGF21ΔN with L-39F7 resulted in WNT/β-catenin signaling in Huh7 cells, a liver cell line that expresses the bridging receptor βKlotho (*KLB*), but not in HEK293 cells, where βKlotho is not expressed (*Figure 3A and G*). This signaling depends on the presence of both FZD- and LRP-binding arms and the ability to bind the bridging receptor, as the removal of the LRP-binding arm L from L-39F7 or inactivation of βKlotho-binding arms (use of FGF21ΔC or FGF21ΔNΔC, or the replacement of 39F7 with αGFP) removes activity in both cells (*Figure 3A–F*). We further examined the target cell-selective WNT signal activation in a two-layer cell culture system as depicted in *Figure 3H*. In this co-culture system, two different cell types are separated by a permeable membrane, and the two different cell lines are stimulated with the same treatment medium in the same vessel. The combination of F-FGF21ΔN and L-39F7 showed a robust increase in WNT/β-catenin signaling in Huh7 cells as compared to the negative control mixture of F-FGF21ΔNΔC and L-39F7. The target cell-specific WNT/β-catenin signaling activation was not detected in HEK293 cells (*Figure 3H*).

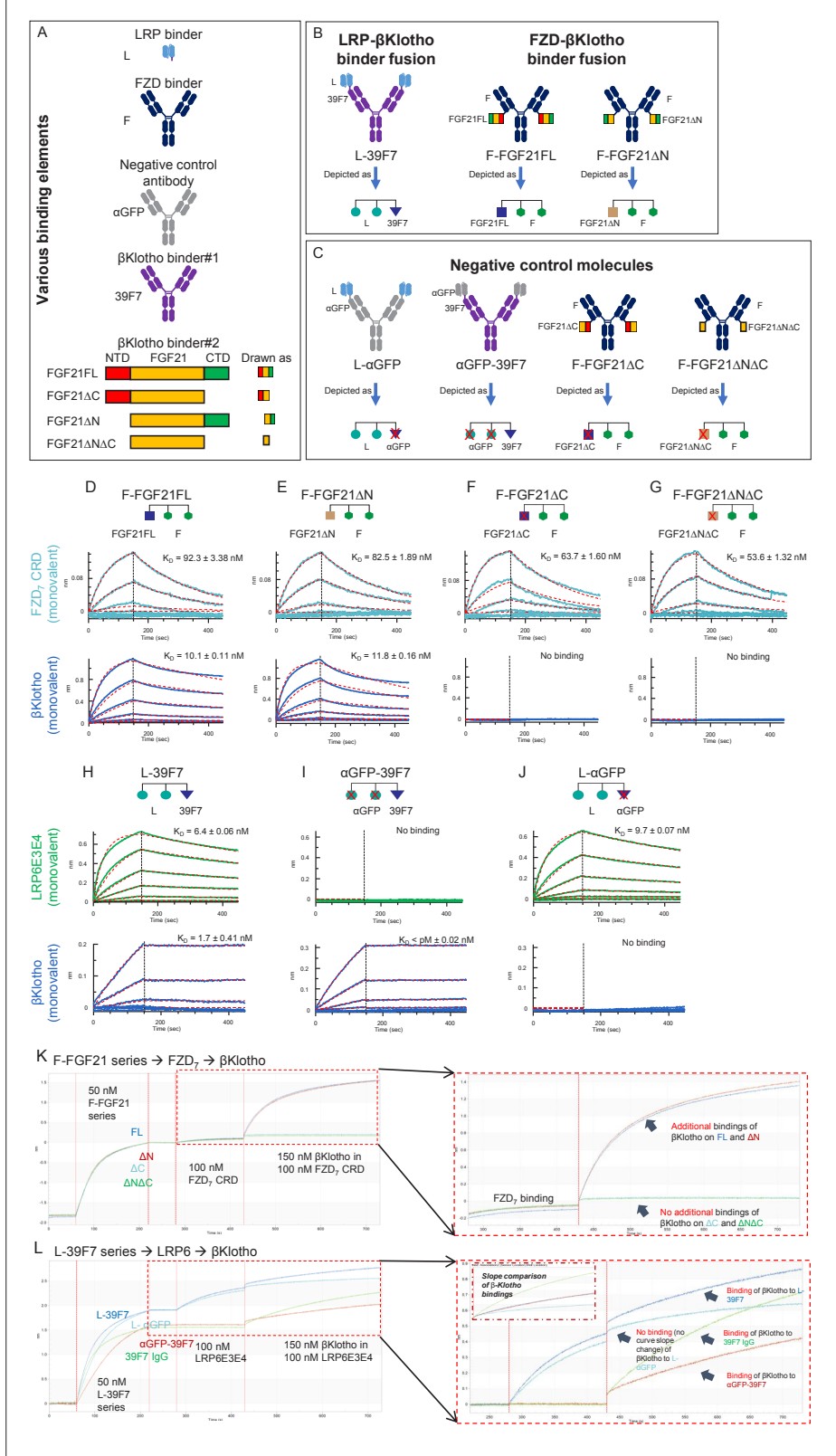

**Figure 2.** Diagrams of the molecules used in the experiment to provide proof of concept for SWIFT. (**A**) Elements that are used in the split molecules. The LRP6 binder element is in scFv format; frizzled (FZD) binder is in IgG1 format; both αGFP IgG1 and αGFP scFv are used for assembly of the negative control molecules. Two types of βKlotho binders are used. Binder#1, 39F7 IgG1, a βKlotho monoclonal antibody; Binder#2, FGF21FL and different

*Figure 2 continued on next page*

*Figure 2 continued*

deletion variants. (**B**) Assembled SWIFT molecules. (**C**) Assembled negative control molecules. (**D–G**) Bindings of various F-FGF21 fusion proteins to FZD$_7$ and βKlotho. Bindings of FZD$_7$ and β-Klotho to F-FGF21FL (**D**), F-FGF21ΔN (**E**), F-FGF21ΔC (**F**), or F-FGF21ΔNΔC (**G**) were determined by Octet. (**H–J**) Binding of LRP6 and βKlotho to L-39F7 (**H**), αGFP-39F7 (**I**), or L-αGFP (**J**) were determined by Octet. Mean K$_D$ values were calculated for the binding curves with global fits (red dotted lines) using a 1:1 Langmuir binding model. (**K**) Step bindings of FZD$_7$ and βKlotho to various F-FGF21 fusion proteins. Sequential binding of F-FGF21FL (blue sensorgram), F-FGF21ΔN (red sensorgram), F-FGF21ΔC (light-blue sensorgram), or F-FGF21ΔNΔC (green sensorgram), followed by FZD$_7$ CRD, then followed by addition of βKlotho on Octet shows simultaneous engagement of both FZD$_7$ and βKlotho to the indicated F-FGF21 proteins. Sensorgrams for FZD$_7$ and βKlotho area (red dotted box) are enlarged at the right. (**L**) Step binding of LRP6 and βKlotho to L-39F7 and its control proteins. Sequential binding L-39F7 (blue sensorgram), L-αGFP (light-blue sensorgram), αGFP-39F7 (red sensorgram), or 39F7 IgG (green sensorgram), followed by LRP6E3E4, then followed by addition of βKlotho on Octet shows simultaneous engagement of both LRP6 and βKlotho to the indicated L-39F7 and its control proteins. Sensorgrams for LRP6 and βKlotho area (red dotted box) are enlarged on the right.

The online version of this article includes the following source data for figure 2:

**Source data 1.**

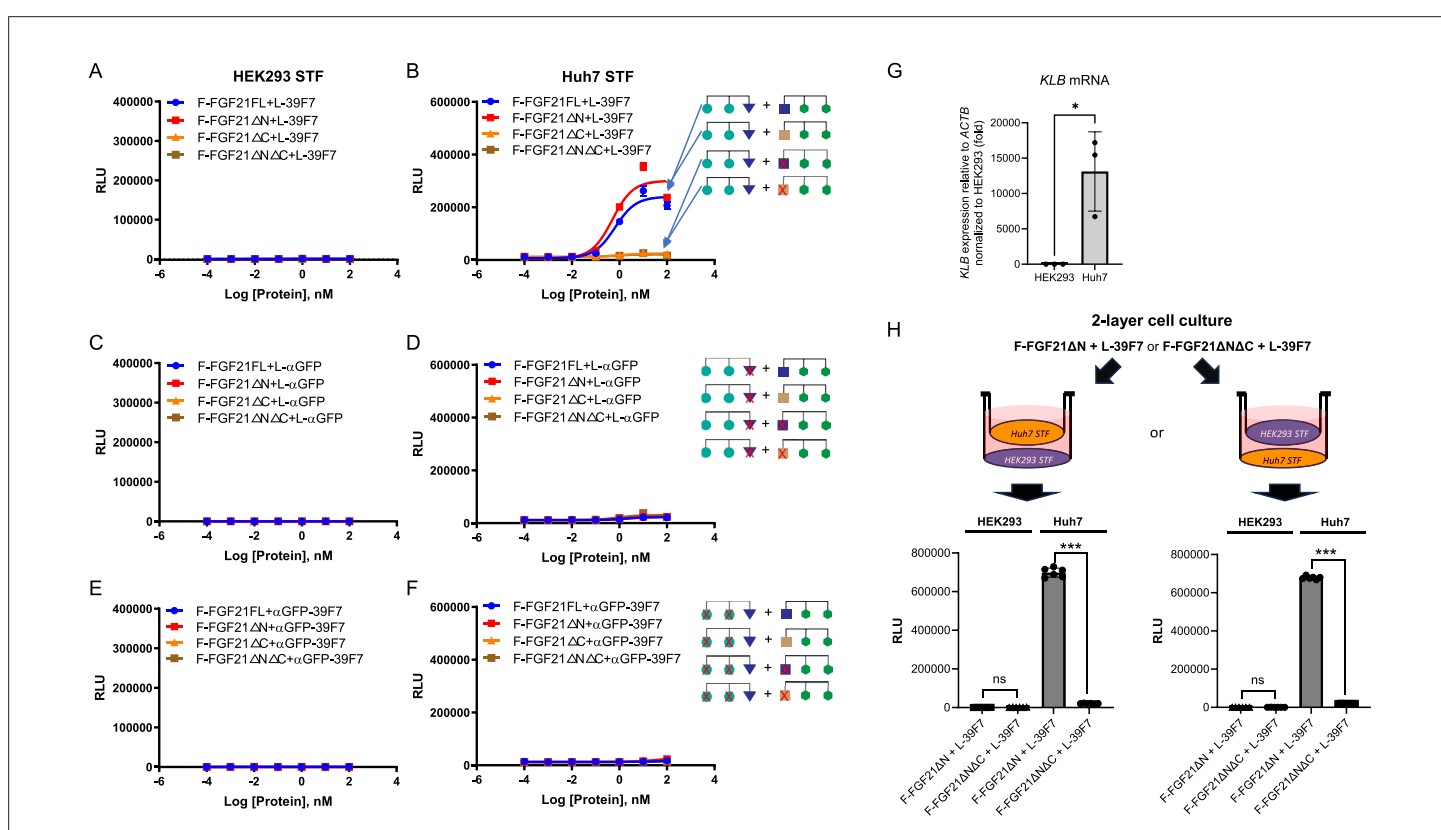

**Figure 3.** Dose-dependent Super TOP-FLASH (STF) assay of SWIFT molecules in HEK293 and Huh7 cells. (**A–F**) Various F-FGF21 fusion proteins in the presence of L-39F7 in HEK293 STF cells (**A**) and Huh7 STF cells (**B**); various F-FGF21 fusion proteins in the presence of L-αGFP in HEK293 STF cells (**C**) and Huh7 STF cells (**D**); and various F-FGF21 fusion proteins in the presence of αGFP-39F7 in HEK293 STF cells (**E**) and Huh7 STF cells (**F**). (**G**) Expression of bridging receptor βKlotho (*KLB*) in HEK293 and Huh7 cells normalized to *ACTB*, relative to HEK293 expression. (**H**) STF responses in a two-layer cell culture system after 16 hr treatment with 10 nM of indicated molecules. Data are representative of three independent experiments performed in triplicate and are shown as mean ± SD. *p<0.05, ***p≤0.001 (unpaired *t*-test).

The online version of this article includes the following source data for figure 3:

**Source data 1.**

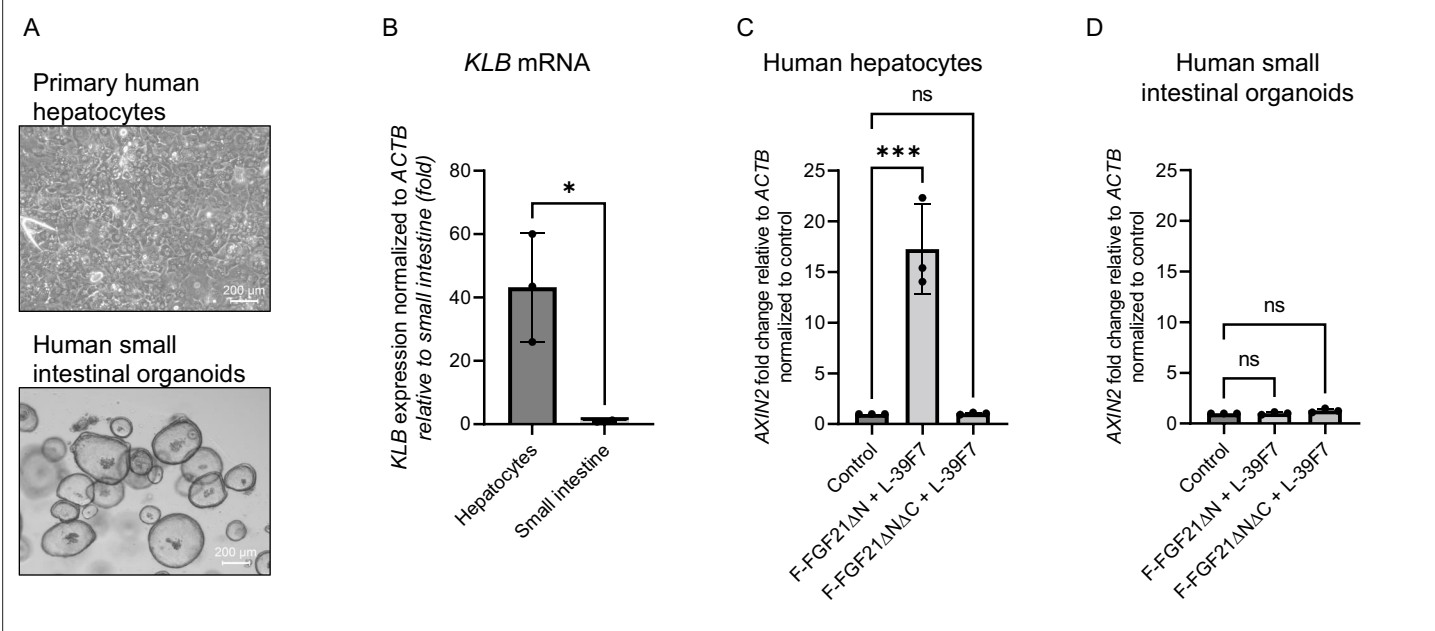

**Figure 4.** Activity of targeted molecules in primary human cells. (**A**) Representative images of primary human hepatocyte cultures in 2D or human small intestinal organoids. Scale bar: 200 µm. (**B**) Expression of bridging receptor βKlotho (*KLB*) in hepatocytes or small intestinal cells. (**C**) WNT target gene *AXIN2* expression normalized to control treatment after 24 hr treatment with 10 nM of the indicated molecules in human hepatocytes. (**D**) WNT target gene *AXIN2* expression normalized to control treatment after 24 hr treatment with 10 nM of the indicated molecules in human small intestinal organoids. *p<0.05 (unpaired *t*-test), ***p≤0.001 (one-way ANOVA), each data point represents an independent experiment performed in duplicate.

The online version of this article includes the following source data for figure 4:

**Source data 1.**

Plating cells in the upper or lower level of the co-culture system did not affect specificity or potency. These results provide experimental evidence supporting the SWIFT concept presented in *Figure 1B*.

## Validation of liver-specific targeting with primary human cells

To test the potency of these molecules in cells directly derived from the tissue of interest, we selected primary human hepatocytes and human small intestinal organoids (*Figure 4A*). These cultures better represent the physiological and transcriptional characteristics of the liver and intestine compared to cell lines. As expected, primary human hepatocytes express bridging receptor βKlotho, while human small intestinal cells do not (*Figure 4B*). The combination of F-FGF21ΔN and L-39F7 in human hepatocytes causes a significant increase in the expression of WNT target gene *AXIN2* after 24 hr compared with the control (*Figure 4C*). The combination with the negative control F-FGF21ΔNΔC does not upregulate *AXIN2*. The same panel of molecules does not elicit any target gene expression in human small intestinal cells, confirming the need for bridging receptor expression for activity (*Figure 4D*). These results in primary cells support the feasibility of this cell-specific targeting strategy, and we expected them to be predictive of tissue specific in vivo responses.

## Discussion

A major challenge in the development of new therapeutics is the effective targeting of drugs toward a desired cell type. Due to the pleiotropic expression and actions of most drug targets in many cell types/organs and the lack of differentiation between diseased vs. normal cells, many drugs exhibit either dose-limiting toxicity or act on multiple cell types of opposing activity, rendering them less effective. Therefore, developing effective cell-targeting methods would allow reduced systemic toxicity and higher efficacy for more effective treatments.

Herein, we have described a novel cell-targeting approach termed BRAID whereby an active drug molecule is divided into inactive parts that are assembled via a bridging receptor specific to the target cell. Although complementation and splitting a molecule into inactive parts have been

explored previously, the novelty of our approach is that there is no requirement for the two split inactive components to interact with one another. Therefore, it does not require cooperative interactions between the split components and receptors for activity. In our approach, assembly of the signaling complex is achieved by simultaneous binding of the two divided inactive components through a common bridging receptor (*Figure 1B*).

We tested this concept using the WNT/β-catenin signaling pathway as a model system. This signaling system requires two receptors, FZD and LRP. The present authors (and others) have generated antibody-based WNT mimetic molecules by linking FZD binders and LRP binders in a single molecule to activate the FZD/LRP receptor complex (*Chen et al., 2020*; *Janda et al., 2017*; *Tao et al., 2019*). To explore the BRAID concept with the WNT signaling receptors, we first divided a tetravalent bispecific WNT mimetic into two components, one containing two FZD-binding domains and the other containing two LRP-binding domains. To each of these two inactive components, a moiety that binds to the bridging receptor, in this case βKlotho (predominantly expressed in hepatocytes), was attached. The two bridging receptor-binding moieties are FGF21 and 39F7, which bind to non-overlapping regions on βKlotho (*Min et al., 2018*). As demonstrated both on WNT-responsive reporter cell lines as well as on primary human hepatocytes and human intestinal organoids, target-cell-specific WNT signaling is indeed observed when the two inactive components are combined. The signaling only occurs on the target-cell hepatocytes and not on HEK293 nor intestinal cells. Furthermore, activity depends on the presence of βKlotho-binding domains on the two inactive components, as removal of βKlotho-binding ability from either the FZD or the LRP half of the molecule (or having βKlotho but not LRP binding) resulted in inactive combinations (*Figures 3 and 4*).

These results provided compelling evidence for this novel cell-targeting concept. We envision that other potential ways of dividing the WNT or WNT-mimetic molecules, including different geometries, FZD/LRP stoichiometry ratios, and linker lengths, or having one FZD- and LRP-binding arm in each component, may also result in active targeted WNT mimetics. We have termed this cell-targeted WNT system SWIFT. Our studies here focused on the WNT signaling pathway, and βKlotho/FGF21/39F7 receptor ligand system was used to illustrate the BRAID/SWIFT cell-targeting concept. Whether these molecules may additionally modulate endocrine FGF signaling and metabolic homeostasis could be the subject of future studies.

WNT plays important roles in many tissues during development as well as in adult tissue homeostasis and injury repair, and dysregulated WNT signaling results in various human disease conditions (*Nusse and Clevers, 2017*). Accordingly, SWIFT offers the opportunity to specifically target WNT activation in desired cell types, facilitating both academic research and the development of novel therapeutics.

In conclusion, we have described a novel cell-targeting approach that may be broadly applied to various signaling systems. This approach, in principle, also allows action upon a combination of different signaling pathways simultaneously, potentially providing additive or synergistic effects.

## Materials and methods
### Key resources table

| Reagent type (species) or resource | Designation | Source or reference | Identifiers | Additional information |
| --- | --- | --- | --- | --- |
| Cell line (*Homo sapiens*) | Expi293F cells | Thermo Fisher Scientific | A14527 | |
| Cell line (*H. sapiens*) | HEK293 STF | https://doi.org/10.1371/journal.pone.0009370 | | Cells containing a luciferase gene controlled by a WNT-responsive promoter |
| Cell line (*H. sapiens*) | Huh7 STF | https://doi.org/10.1038/s41598-020-70912-3 | | Cells containing a luciferase gene controlled by a WNT-responsive promoter |

*Continued on next page*

*Continued*

| Reagent type (species) or resource | Designation | Source or reference | Identifiers | Additional information |
|---|---|---|---|---|
| Biological sample (*H. sapiens*) | Primary hepatocytes | BioIVT | X008001-P | 10-Donor Human Cryoplateable |
| Biological sample (*H. sapiens*) | Human small intestinal organoids | Kuo Lab Stanford | NA | Primary cell derived 3D organoid culture |
| Peptide, recombinant protein | Human βKlotho | Fisher Scientific | Cat# 5889KB-050 | |
| Peptide, recombinant protein | Human FZD$_7$ CRD | https://doi.org/10.1038/s41467-021-23374-8 | | Produced in Expi293F cells |
| Peptide, recombinant protein | Human LRP6 E3E4 | https://doi.org/10.1016/j.chembiol.2020.02.009 | | Produced in Expi293F cells |
| Peptide, recombinant protein | Fc-R-spondin 2 | https://doi.org/10.1038/s41598-020-70912-3 | | Produced in Expi293F cells |
| Peptide, recombinant protein | Recombinant Human EGF | PeproTech | Cat# AF-100-15 | |
| Peptide, recombinant protein | Recombinant Human Noggin | PeproTech | Cat# 120-10C | |
| Peptide, recombinant protein | Human Gastrin I | Tocris | Cat# 30061 | |
| Commercial assay or kit | Luciferase Assay System | Promega | E1501 | |
| Chemical compound, drug | IWP2 | Tocris Bioscience | Cat# 3533 | Porcupine inhibitor |
| Chemical compound, drug | B27 | Thermo Scientific | Cat# 17504044 | |
| Chemical compound, drug | N2 | Thermo Scientific | Cat# 17502048 | |
| Chemical compound, drug | *N*-Acetylcysteine | Sigma-Aldrich | Cat# A9165 | |
| Chemical compound, drug | Nicotinamide | Sigma-Aldrich | Cat# N0636 | |
| Chemical compound, drug | A83-01 | Tocris | Cat# 2939 | |
| Chemical compound, drug | SB202190 | Tocris | Cat# 126410 | |
| Software, algorithm | Octet Data Analysis 9.0 | Sartorius | https://www.sartorius.com/en/products/protein-analysis/octet-bli-detection/octet-systems-software | |
| Software, algorithm | Prism | GraphPad | https://www.graphpad.com/scientific-software/prism/ | |
| Other | Matrigel Matrix | Corning | CB40230C | Extracellular matrix for 3D organoid growth and plate coating |

## Cell lines

Expi293F cells were grown in Expi293 Expression Medium (Thermo Fisher Scientific) at 37°C with humidified atmosphere of 8% $CO_2$ in air on an orbital shaker. HEK293 STF cells were maintained in Dulbecco's modified Eagle's medium (DMEM) supplemented with 10% fetal bovine serum (Fisher Scientific) at 37°C in a 5% $CO_2$ environment. Huh7 STF cells were maintained in DMEM supplemented with 10% fetal bovine serum and non-essential amino acids (Fisher Scientific) at 37°C in a 5% $CO_2$

environment. Cell line authentications were performed by the vendor or the source, mycoplasma testing by PCR Mycoplasma detection kit.

## Molecular cloning

All constructs were cloned into pcDNA3.1(+) mammalian expression vector (Thermo Fisher). The L-scFv binder was constructed by fusing the heavy-chain variable region (VH) to the N-terminus of the light-chain variable (VL) region with a 15-mer linker (GSAASGSSGGSSSGA). The anti-GFP scFv binder was constructed by fusing the VH to the N-terminus of the VL with a 15-mer-linker (GGGGSGGGGSGGGGS). All human IgG1 constructs contain the L234A/L235A/P329G mutations (LALAPG) in Fc domain to eliminate effector function (*Lo et al., 2017*). For generating the constructs of scFv-39F7 IgG1, the scFv binder was fused to the N-terminus of 39F7 LC with a 5-mer-linker GSGSG. To generate the construct of L-αGFP IgG1, the L-scFv was fused to the N-terminus of αGFP LC with a 15-mer-linker GSGSGGSGS-GGSSGG. To generate the FGF21-variant-appended F IgG molecules, FGF21 variants were fused to the C-terminus of the LC of F IgG1 (LALAPG) with a 5-mer-linker (GGSGS). FGF21FL has the mature protein sequence (H29-S209) with RGE mutations in the C-terminus (*Stanislaus et al., 2017*); FGF21ΔN is the sequence of FGF21FL with the deletion of the N-terminus H29–R45; FGF21ΔC is the sequence of FGF21FL with the deletion of the C-terminus S190–S209; FGF21ΔNΔC is the sequence of FGF21FL with the deletion of both the N-terminal H29–R45 and the C-terminal S190–S209.

## Protein production

All recombinant proteins were produced in Expi293F cells (Thermo Fisher Scientific) by transient transfection. The proteins were first purified using CaptivA Protein A affinity resin (Repligen), unless otherwise specified. All proteins were further purified with Superdex 200 Increase 10/300 GL (GE Healthcare Life Sciences) size-exclusion chromatography (SEC) using 1× HBS buffer (20 mM HEPES pH 7.4, 150 mM NaCl). The proteins were subsequently examined by SDS-polyacrylamide electrophoresis and estimated to have >90% purity.

## STF assays

WNT signaling activity was measured using HEK293 and Huh7 cells containing a luciferase gene controlled by a WNT-responsive promoter (STF assay) as previously reported by *Zhang et al., 2020*. In brief, cells were seeded at a density of 10,000 per well in 96-well plates. For two-layer cell culture, either Huh7 STF or HEK293 STF cells were seeded into tissue culture-treated 6.5-mm transwell inserts (top well, Corning) with the density of 20,000. The transwell inserts were then placed into wells in a 24-well plate where HEK293 STF or Huh7 STF cells were plated with the density of 116,000 (bottom well), respectively. Then, 24 hr later, indicated WNT signaling activators were added in the presence of 3 μM IWP2 to inhibit the production of endogenous WNTs and the presence of 20 nM Fc-R-Spodin 2. Cells were lysed with Luciferase Cell Culture Lysis Reagent (Promega), and luciferase activity was measured with Luciferase Assay System (Promega) using vendor-suggested procedures.

## Affinity measurement and step-binding assay

Binding kinetics of F-FGF21 series (F-FGF21FL, F-FGF21ΔN, F-FGF21ΔC, and F-FGF21ΔNΔC) to human FZD$_7$ CRD and βKlotho (Fisher Scientific) or L-39F7 series (L-39F7, αGFP-39F7, and L-αGFP) to human LRP6E3E4 and βKlotho, respectively, were determined by bio-layer interferometry (BLI) using an Octet Red 96 (PALL ForteBio) instrument at 30°C and 1000 rpm with AHC biosensors (Sartorius). Various F-FGF21 or L-39F7 proteins were diluted to 50 nM in the running buffer and captured by the AHC biosensor, followed by dipping into wells containing the FZD$_7$ CRD, LRP6E3E4, and βKlotho at different concentrations in a running buffer or into a well with only the running buffer as a reference channel. The dissociation of the interaction was followed with the running buffer. The monovalent K$_D$ for each binder was calculated using Octet System software, based on fitting to a 1:1 binding model.

Step-binding assays were performed with the BLI using the Octet Red 96 instrument at 30°C and 1000 rpm with AHC biosensors. Various F-FGF21 or L-39F7 proteins were diluted to 50 nM in the running buffer and captured by the AHC biosensor, followed by dipping into wells containing the 100 nM FZD$_7$ CRD or 100 nM LRP6E3E4. The sensor chips next moved into 150 nM βKlothos containing 100 nM FZD$_7$ CRD or containing 100 nM LRP6E3E4 to check the additional bindings of β-Klotho. Sensorgram slopes were compared for βKlotho bindings.

### Primary human cells

Human hepatocytes were purchased from BioIVT (10-donor pooled cryoplateable X008001-P) and cultured in LONZA hepatocyte maintenance medium (CC-3198). Briefly, plastic culture plates were coated with 20% Matrigel Matrix (CB40230C) and cells were plated in plating medium (BioIVT Z990003). After 4 hr, the medium was changed to maintenance medium and refreshed every day for 3 d prior to the 24 hr experiment.

Human small intestinal organoids were kindly supplied by the Calvin Kuo Lab at Stanford. Organoids were maintained and expanded as previously described (*Sato et al., 2011*). Briefly, adapted expansion medium contained advanced DMEM, 10 mM HEPES, 1× GlutaMAX, 1× penicillin–streptomycin, 1× B27, 1× N2, 1.25 mM *N*-acetylcysteine, 10 mM nicotinamide, 50 ng/mL recombinant human EGF, 50 ng/mL recombinant human Noggin, 20 nM R-Spondin 2, 0.1 nM L-F Wnt mimetic, 10 nM recombinant gastrin, 500 nM A83-01, and 10 μM SB202190.

Treatment of molecules was done in the presence of 20 nM R-Spondin 2 for 24 hr at a concentration of 10 nM. After 24 hr, the cells were harvested and RNA collected for quantitative polymerase chain reaction (qPCR). Each experiment with both primary human hepatocytes and human small intestinal organoids was repeated three times.

### qPCR analysis of gene expression

RNAs from HEK293, Huh7 cells, or primary human cells were extracted using the QIAGEN RNeasy Micro Kit (QIAGEN). cDNA was produced using the SuperScript IV VILO cDNA Synthesis Kit (Thermo Fisher). βKlotho (*KLB*) RNA was quantified using Maxima SYBR Green qPCR master mix on a Bio-Rad CFX96 real-time PCR machine.

Cycle threshold ($C_t$) values were normalized to the expression of constitutive *ACTINB* RNA using the following oligomers: ACTB_F1: CTGGAACGGTGAAGGTGACA. ACTB_R1: AAGGGACTTCCT GTAACAATGCA. KLB_F1: ATCTAGTGGCTTGGCATGGG. KLB_R1:CCAAACTTTCGAGTGAGCCTTG. KLB_F2:CACTGAATCTGTTCTTAAGCCCG. KLB_R2: GGCGTTCCACACGTACAGA. KLB_F3: GGAG GTGCTGAAAGCATACCT. KLB_R3: TCTCTTCAGCCAGTTTGAATGC.

### Materials availability statement

All unique/stable reagents generated in this study are available with a completed Materials Transfer Agreement.

## Acknowledgements

We thank Leona Cheng, Haili Zhang, Jasmine Tan, Hayoung Go, and Sean Bell for technical support and discussions. We also thank Huy Nguyen for the illustration in *Figure 1*. We thank Wen-Chen Yeh and Craig Parker for critical reading of the manuscript and Kathee Littrell for editorial support.

## Additional information

### Competing interests

Hui Chen, Sung-Jin Lee, Nicholas Suen, Timothy T Suen, Chenggang Lu, Yorick Post: The authors are current full-time employees and shareholders of Surrozen, Inc. Ryan Li, Asmiti Sura, Archana Dilip, Yan Pomogov, Meghah Vuppalapaty: The authors were former full-time employees and shareholders of Surrozen, Inc. Yang Li: The author is a current full-time employee and shareholder of Surrozen, Inc. YL is Executive Vice President of Research at Surrozen, Inc.

### Funding

| Funder | Grant reference number | Author |
|--------|------------------------|--------|
| Surrozen, Inc | | Hui Chen |

The funders had no role in study design, data collection and interpretation, or the decision to submit the work for publication.

## Author contributions
Hui Chen, Sung-Jin Lee, Yorick Post, Resources, Formal analysis, Supervision, Investigation, Visualization, Methodology, Writing – original draft, Writing – review and editing; Ryan Li, Asmiti Sura, Nicholas Suen, Archana Dilip, Yan Pomogov, Meghah Vuppalapaty, Timothy T Suen, Chenggang Lu, Resources, Investigation, Visualization, Methodology, Writing – review and editing; Yang Li, Conceptualization, Resources, Formal analysis, Supervision, Funding acquisition, Visualization, Methodology, Writing – original draft, Writing – review and editing

## Author ORCIDs
Yang Li https://orcid.org/0000-0002-7134-5685

Reviewer #1 (Public Review): https://doi.org/10.7554/eLife.90221.3.sa1
Reviewer #2 (Public Review): https://doi.org/10.7554/eLife.90221.3.sa2
Author Response https://doi.org/10.7554/eLife.90221.3.sa3

## Additional files

### Supplementary files
• MDAR checklist

### Data availability
All data generated or analysed during this study are included in the manuscript and supporting file; Source Data files have been provided for Figures 2, 3, and 4.

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
