## [Editor Report · eLife assessment]

This **important** study presents a new way to selectively activate a cell signaling pathway in a specific cell type by designer ligands that link signaling co-receptors to a marker specific to the target cells. **Convincing** experimental results demonstrate that the agonist molecules activate Wnt signaling in target cells expressing the marker as intended. More broadly, this concept could be used to induce Wnt signaling or another pathway initiated by co-receptor association in a cell type-specific manner. In vitro results in this study could be further strengthened by assessing the biological consequences of Wnt activation in target cells.

---

## [Referee Report · Reviewer #1 (Public Review)]

Summary: The goal of this study was to develop and validate novel molecules to selectively activate a cell signaling pathway, the Wnt pathway in this case, in target cells expressing a specific receptor. This was achieved through a two-component system that the authors call BRAID, where each component simultaneously binds the target cell-specific marker BKlotho and a Wnt co-receptor. These components, called SWIFT molecules, bring together the Wnt co-receptors LRP and FZD, activating the pathway specifically in cells that express BKlotho.

Results presented in the study demonstrate the desired activity of SWIFT molecules; the binding assays support simultaneous association of SWIFT with BKlotho and a Wnt co-receptor, and the Wnt reporter and qPCR assays support pathway activation in cell lines and primary cells in a BKlotho-dependent manner. In the future, the BRAID approach could be applied to activate Wnt signaling or another pathway initiated by a co-receptor complex in a cell type-specific manner, and/or in a FZD subtype-specific manner to activate distinct branches of Wnt signaling.

Strengths:

• This study successfully demonstrates a novel way to activate Wnt signaling in target cells expressing a specific marker. Given the role of the Wnt signaling pathway in key processes such as cell proliferation and tissue renewal and the value of modulating cell signaling in a cell type-specific manner, the cell targeting system developed here holds great therapeutic and research potential. It will be curious to see whether the BRAID design can be applied to other cell surface markers for Wnt activation, or for activation of other signaling pathways that require co-receptor association.

• Octet assay results show simultaneous binding of SWIFT molecules to both the Wnt co-receptor FZD/LRP and BKlotho, while negative control molecules without the FZD/LRP or BKlotho-binding module show neither receptor binding nor Wnt pathway activation. These results indicate that SWIFT molecules function through the intended mechanism.

• Exposure of two cell types simultaneously exposed to the SWIFT molecules in 2-layer cell culture demonstrated the ability of the molecules to activate Wnt signaling in a cell type- and BKlotho expression-specific manner.

Weaknesses:

• The study does not address whether the targeted cells express FGFR1c/2c/3c and whether the FGF21 full length moiety or the 39F7 IgG moiety of SWIFT molecules could unintentionally activate FGF signaling in these cells.

---

## [Referee Report · Reviewer #2 (Public Review)]

Summary:

The study introduces BRAID, a novel approach for targeting drugs to specific cell types, addressing the challenges of pleiotropic drug actions. Unlike existing methods, this one involves breaking a protein drug molecule into inactive parts that are then put back together using a bridging receptor on the target cell. The individual components of this assembly are not required to be together, thereby affording it a degree of flexibility. The authors applied this idea to the WNT/-catenin signaling pathway by splitting a WNT mimic into two parts with FZD and LRP binding domains and bridging receptors. This combined method, which is called SWIFT, showed that WNT signaling was turned on in target cells, showing that cell-specific targeting is. The technique shows promise for the development of therapeutics, as it provides a way to more precisely target signaling pathways.

The authors have effectively elucidated their strategy through visually appealing diagrams, providing clear and thorough visual aids that facilitate comprehension of the concept. In addition, the authors have provided convincing evidence that the C-terminal region of FGF21 is essential for the binding process. Their meticulous and thorough presentation of experimental results emphasizes the significance of this specific binding domain and validates their findings.

Strengths:

BRAID, a novel cell targeting method, divides an active drug molecule into inactive components formed by a bridging receptor. This novel approach to cell-specific drug action may reduce systemic toxicity.

The SWIFT approach successfully targets cells in the WNT/β-catenin signaling pathway. The approach activates WNT signaling only in target cells (hepatocytes), proving its specificity.

The study indicates that the BRAID approach can target various signaling systems beyond WNT/β-catenin, indicating its versatility. Therapeutic development may benefit from this adaptability.

Weaknesses:

The study shows the SWIFT approach works in vitro using cell lines, primary human hepatocytes, and human intestinal organoids, but it lacks in vivo animal model or clinical validation. I believe future studies will determine this aspect.

The success of SWIFT depends on the presence and expression of the bridging receptor (βKlotho) on target cells. The approach may fail if the target receptor is not expressed.

---

## [Author Response]

The following is the authors’ response to the original reviews.

**Public Reviews:**

**Reviewer #1 (Public Review):**
Weaknesses:Reviewer comment: Here, the activity of SWIFT molecules was assessed in single cell types with or without BKlotho expression. Ultimately, the ability of the SWIFT molecules to activate Wnt signaling in a cell type-specific manner should be tested in the context of many different cellular identities that express BKlotho to different extents. It would be good to demonstrate that Wnt activation by SWIFT correlates with BKlotho expression level in multiple cell types - such data would strengthen the claim of cell-type specificity.

Response: We agree with the reviewer’s comment, it would be interesting to correlate the signaling level to the expression levels of βKlotho. The tools to carry out such an experiment are not currently available, as this would require a culture system that allows efficient growth of different cell types, and the reagents to detect both the receptor protein levels of βKlotho (as well as FZD/LRP) and signaling levels. We did perform an additional experiment to further support this targeting approach using a 2-layered (transwell) cell culture system. In this culture system, one cell type is put into the top well and the other cell type is put into the bottom well. Molecules to be tested were added to the media which is shared and freely diffuse across the two cell types. In this 2-layer cell system, the results again demonstrate the ability of the SWIFT molecules to specifically induce signaling only in βKlotho expressing hepatoma Huh7 cells and not in non-targeting HEK293 cells. This new data is included as Fig. 3H in the revised manuscript.

Reviewer comment: The study does not address whether the targeted cells express FGFR1c/2c/3c and whether the FGF21 full-length moiety or the 39F7 IgG moiety of SWIFT molecules could unintentionally activate FGF signaling in these cells.

Response: We agree with the reviewer’s comment. The receptor βKlotho and its binders (FGF21 and 39F7) were used to test the BRAID/SWIFT concept, the effects on FGF signaling were not the focus of the current study. This comment has now been added to the revised manuscript in the discussion. Inclusion of αGFP controls in the study also suggests the observed reporter activity in the targeted scenario is unlikely caused indirectly by any unexpected FGF signaling.

**Reviewer #2 (Public Review):**
Weaknesses:Reviewer comment: The study shows the SWIFT approach works in vitro using cell lines, primary human hepatocytes, and human intestinal organoids, but it lacks an in vivo animal model or clinical validation. The applicability of this approach to therapy is still unknown.

Response: The βKlotho binder, 39F7, is specific to the human receptor and does not cross react with mouse. Unfortunately, we are not able to test these SWIFTs in a mouse model.

Reviewer comment: The success of SWIFT depends on the presence and expression of the bridging receptor (βKlotho) on target cells. The approach may fail if the target receptor is not expressed or available.

Response: We agree with the reviewer, the SWIFT molecules should not induce signaling on cells where bridging receptor is not expressed, therefore, achieving target cell specificity. As pointed out by the reviewer, finding the right bridging receptor on the target cell is critical.

**Reviewer #1 (Recommendations For The Authors):**
Reviewer comment 1: One way to further validate the specificity of SWIFT molecules would be to apply them to a mix of different cell types and quantify BKlotho level and Wnt reporter activity at the single cell level, potentially through imaging, FACS, or transcriptomics.

Response: We agree with the reviewer’s comment, it would be interesting to correlate the signaling level to the expression levels of βKlotho. The tools to carry out such an experiment are not currently available, as this would require a culture system that allows efficient growth of different cell types, and the reagents to detect both the receptor protein levels of βKlotho (as well as FZD/LRP) and signaling levels. We did perform an additional experiment to further support this targeting approach using a 2-layered (transwell) cell culture system. In this culture system, one cell type is put into the top well and the other cell type is put into the bottom well. Molecules to be tested were added to the media which is shared and freely diffuse across the two cell types. In this 2-layer cell system, the results again demonstrate the ability of the SWIFT molecules to specifically induce signaling only in βKlotho expressing hepatoma Huh7 cells and not in non-targeting HEK293 cells. This new data is included as Fig. 3H in the revised manuscript.

Reviewer comment 2: The experiments presented demonstrate activation of one signaling pathway in cells specifically expressing a target receptor rather than demonstrating "the feasibility of combining different signaling pathways" as stated in the abstract.

Response: We thank the reviewer for pointing this out and have adjusted the sentence accordingly.

Reviewer comment 3: What are the biological consequences of activating Wnt signaling in cells expressing BKlotho and why is that of interest? Could these biological outcomes be used as an additional, perhaps more consequential, readout for SWIFT activity?

Response: βKlotho is expressed on several different cell types that include hepatocytes, WAT, BAT, and certain regions in CNS. Our studies here focused on the WNT signaling pathway, and βKlotho/FGF21/39F7 receptor ligand system was used to illustrate the BRAID/SWIFT cell targeting concept. Whether these molecules may additional modulate endocrine FGF signaling and metabolic homeostasis, and whether there is any interaction between βKlotho and Wnt signaling pathways could be the subject of future studies. This is now added to the revised manuscript.

Reviewer comment 4: The manuscript would benefit from a careful review to improve wording and address grammatical errors.

Response: We thank the reviewer for this suggestion, and we have now had another round of language editing by a professional service.

**Reviewer #2 (Recommendations For The Authors):**
Reviewer comment 1. The expression of KLB in Fig 3G and 4B seems way too low and may not represent the amount on the cell surface. Did the authors validate the expression on the cell surface?

Response: In both figures we have displayed the expression level normalized to housekeeping gene ACTB. Housekeeping genes such as ACTB can be among the most abundant transcripts in a cell. The observation that KLB mRNA detection is below ACTB mRNA levels is expected and we would argue not too low. The average real-time PCR cycle threshold (Ct) for KLB in Huh7 and primary hepatocytes was 18 and 24 respectively. To avoid any confusion, we have now displayed the expression data normalized to HEK293 and intestinal organoids as a fold difference in a new Figure 3G and 4B.

Comment 2. Fig 3G needs statistical significance.

Response: We thank the reviewer for highlighting this, we have now included the statistical analysis in an updated Figure 3G.